# Evaluation of deep convolutional neural networks for in situ hybridization gene expression image representation

Pegah Abed-Esfahani[1], Benjamin C. Darwin[2], Derek Howard[1], Nick Wang[2], Ethan Kim[1,3], Jason Lerch[2,4], Leon French[1,3,5,6]*

1 Krembil Centre for Neuroinformatics, Centre for Addiction and Mental Health (CAMH), Toronto, Canada, 2 Mouse Imaging Centre, Hospital for Sick Children, Toronto, Canada, 3 Institute for Medical Science, University of Toronto, Toronto, Canada, 4 Wellcome Centre for Integrative Neuroimaging, University of Oxford, Oxford, United Kingdom, 5 Campbell Family Mental Health Research Institute, Centre for Addiction and Mental Health, Toronto, Canada, 6 Department of Psychiatry, University of Toronto, Toronto, Canada

* leonfrench@gmail.com

**Data Availability Statement:** Code, supplementary tables, embeddings, and data files for reproducing the analyses are publicly available online at https://figshare.com/s/43ebba2711adc3ccdc13, and

## Abstract

High resolution *in situ* hybridization (ISH) images of the brain capture spatial gene expression at cellular resolution. These spatial profiles are key to understanding brain organization at the molecular level. Previously, manual qualitative scoring and informatics pipelines have been applied to ISH images to determine expression intensity and pattern. To better capture the complex patterns of gene expression in the human cerebral cortex, we applied a machine learning approach. We propose gene re-identification as a contrastive learning task to compute representations of ISH images. We train our model on an ISH dataset of ~1,000 genes obtained from postmortem samples from 42 individuals. This model reaches a gene re-identification rate of 38.3%, a 13x improvement over random chance. We find that the learned embeddings predict expression intensity and pattern. To test generalization, we generated embeddings in a second dataset that assayed the expression of 78 genes in 53 individuals. In this set of images, 60.2% of genes are re-identified, suggesting the model is robust. Importantly, this dataset assayed expression in individuals diagnosed with schizophrenia. Gene and donor-specific embeddings from the model predict schizophrenia diagnosis at levels similar to that reached with demographic information. Mutations in the most discriminative gene, Sodium Voltage-Gated Channel Beta Subunit 4 (*SCN4B*), may help understand cardiovascular associations with schizophrenia and its treatment. We have publicly released our source code, embeddings, and models to spur further application to spatial transcriptomics. In summary, we propose and evaluate gene re-identification as a machine learning task to represent ISH gene expression images.

## Introduction

Properties of neurons and glia are primarily determined by unique combinations of expressed genes [1, 2]. Over 80% of genes are expressed in the brain, and they often display spatially

https://github.com/PegahA/Human_Brain_ISH_ML.

**Funding:** This study was supported by the CAMH Foundation, McLaughlin Centre, Canada Foundation for Innovation, and a National Science and Engineering Research Council of Canada (NSERC) Discovery Grants to LF. The funders had no role in study design, data collection and analysis, decision to publish, or preparation of the manuscript".

**Competing interests:** The authors have declared that no competing interests exist.

variable patterns at regional and cellular scales [3]. These patterns provide insight into functional circuitry and behaviour that we seek to understand in order to diagnose and treat complex mental illnesses. The Allen Brain Atlases contains terabytes of *in situ* hybridization (ISH) images of the postmortem mouse and human brain [4, 5]. A wealth of spatial gene expression information is contained in these images but has yet to be fully exploited due to their quantity and qualitative nature.

Several computational approaches have been used to quantify expression in neural ISH images. A hand-engineered informatics pipeline was developed by the Allen Institute for Brain Science to extract information from the images [5, 6]. This pipeline was initially developed for the whole mouse brain and later adapted to ISH images of human tissue sections. For the human brain, this pipeline was used to quantify expression across cortical layers for manually chosen cortical patches [4, 7]. Other groups have applied scale-invariant feature transform descriptors (SIFT) [8, 9], deep convolutional neural networks [10, 11], and deep convolutional denoising autoencoders [12] to represent mouse brain images. While limited to the mouse brain, these studies have found that deep convolutional neural networks outperform SIFT based methods.

Deep convolutional neural networks perform best when trained on large amounts of labelled images. Most ISH gene expression datasets of the brain lack a clear target label. However, each image assays the expression of a specific gene. A classification-based approach might not be successful due to the few example images per gene and may limit generalization to unseen genes. In this case, a contrastive loss or metric learning approach is applicable [13]. Gene identity provides a training signal to learn an embedding space where ISH patches of the same gene are closer than patches from two different genes. This approach provides an expanded set of labelled examples as individual images are contrasted using a Siamese neural network or triplet loss function [14–17]. Inspired by progress in person re-identification [18], we propose gene re-identification as a new task to learn representations of gene expression patterns.

Our approach and tools to undertake gene re-identification establish a consistent resource for neuroscience and machine learning researchers. In comparison to person re-identification tasks and datasets that raise privacy concerns [19], images of anonymized brain tissue samples cannot be used to identify subjects. Furthermore, unlike most entities used in re-identification studies, genes are relatively static and allow generalization tests. These attributes provide a stable framework to assess new techniques and discover insights into biological processes.

In this study, we explore transfer learning, contrastive loss, and deep convolutional neural networks to embed gene expression patterns from the human brain. We test if the learned representations contain biologically meaningful information about molecular neuroanatomy at the level of genes. To assess generalizability, we evaluate our model on images obtained from a different brain region and set of individuals. Finally, we test if learned embeddings can discriminate between ISH images obtained from controls and those diagnosed with schizophrenia. Fig 1 provides an overview of our approach that leverages the triplet loss for gene re-identification and transfer learning.

## Methods

### Cortex study dataset

Our primary dataset assayed spatial gene expression of approximately 1,000 genes in the human cerebral cortex [4]. Using colorimetric, digoxigenin-based high-throughput ISH methods, Zeng and colleagues examined expression patterns in the neuropathologically normal brain. They sampled from the visual (V1 and V2) and mid temporal cortices of adult

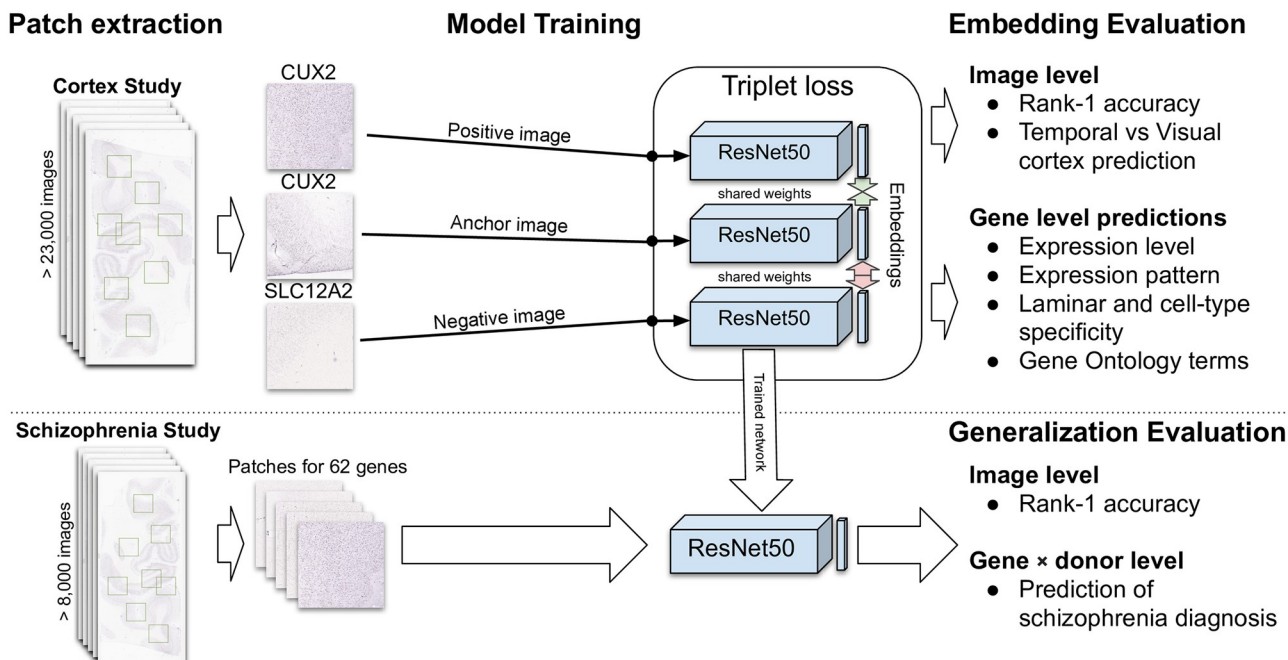

**Fig 1. Overview of this study.** Patches are extracted from ISH images (left) and used to train a single ResNet50 model that shares weights in a triplet loss architecture (middle top). Learned embeddings are then evaluated at the level of genes and images. The trained ResNet model is used to embed patches from a second dataset of ISH images (bottom left). These embeddings are then evaluated at the image level to assess generalizability. Embeddings for individual genes at the level of donors are used to predict schizophrenia diagnosis (bottom right).

individuals without diagnoses of neuropsychiatric disorders. In total, 42 donors were used in this study and 62% were male. The age of the individuals ranged from 22 to 56 years old (40.2 average), and the postmortem interval was 28.8 hours on average (range: 10 to 68 hours).

All images were downloaded via the Allen Brain Institute API with a downsampling rate of 2.5. Before downsampling, the full resolution images captured 1 μm per pixel [4]. Gene level manual qualitative annotations were obtained from S2 Table of [4]. Only annotations that mark five or more genes were used in the classification tasks. Expression level annotations from the V1 cortex were used due to completeness.

## Schizophrenia study dataset

We downloaded images from a second study to further evaluate our model that was trained on the Cortex Study images. Compared to the Cortex Study that was focused on neuroanatomy, this project assayed expression in a smaller set of genes in normal controls and schizophrenic patients [7]. In addition, expression was assayed in the dorsolateral prefrontal cortex due to its associations with schizophrenia and neurodevelopment [20, 21]. Images were obtained using the same procedure as the Cortex Study.

Demographic data were obtained from the Allen Brain Atlas API (diagnosis, donor age, postmortem interval, tissue pH, smoking status, sex and race/ancestry).

## Image preprocessing

In order to train on biologically meaningful images, we used a deep learning-based segmentation pipeline to separate gray and white matter from the background in the full ISH images. This pipeline was built upon fastAI and uses a 34 layer U-net architecture based on a ResNet34

model pre-trained on ImageNet (https://github.com/Mouse-Imaging-Centre/fastCell) [22–24]. We first manually segmented 40 random ISH images from the Cortex Study to provide a training set. These 40 examples are used to train the network on full images downsampled to a resolution of 224x224 pixels (images padded with white to form squares). This network is then used to provide foreground masks for all images.

These coarse foreground masks were then used to filter random patches that are extracted from each full image. Specifically, every patch must contain at least 90% foreground pixels. At most, 50 patches were extracted per image. For over 98% of the images, 50 patches were found within the stopping point of 500 random patch selections. The final patch resolution was 256x256 pixels, which was downsampled from 1024x1024 sized patches in the original image. This patch size in the downloaded image was chosen to fit the human cortex's full width (~2.5mm). This resolution differs from the foreground mask stage due to the different tools used, but we note they are operating at different levels (image versus patch). For both stages, the resolution is constant across the training, test and validation sets.

## Triplet loss

To learn gene embeddings for gene re-identification, we used the triplet loss function. Briefly, triplet loss trains a neural network to minimize the distance between two positive images of the same gene while maximizing the distance to a third image of a different gene. Based on Hermans et al., the mining of hard triplets was set to the default hard strategy and the margin was set to soft. We used an implementation of a triplet loss variant that performs well on person re-identification [18]. The majority of the default settings in the implementation by Herman and colleagues were not tuned. We made a minor modification that additionally flipped the input patches vertically (upside down).

A ResNet50 model pre-trained on Large Scale Visual Recognition Challenge 2012 dataset of natural images was used as the base network for the triplet loss network (https://github.com/tensorflow/models/tree/master/research/slim#pre-trained-models) [22, 24]. Embeddings generated by this pre-trained network were also used for baseline comparisons. Guided by Hermans and colleagues, the output vector dimension of this model is 128.

## Experimental setup for model tuning

For model tuning and evaluation, the assayed genes and corresponding patches were split into training (80%), validation (10%) and test sets (10%). While data from the donor brains were distributed across these splits, this design ensured that evaluation was performed on unseen genes. Hyperparameters were chosen based on evaluations on the validation set. Although patches were used for training, the primary metric for tuning was the rank-1 accuracy or matching rate at the level of images (same as rank-1 precision). This metric is the proportion of images for which the closest image in the embedding space assays the expression of the same gene (Euclidean distance metric). This re-identification test excludes images obtained from the same donor as the query image.

When tuning the hyperparameters on the training and validation sets, we experimented with different `flip_augment` (true or false), `batch_p` (number of genes per batch: 5, 10, 12, 14, 16, 17,18, 20, 40, 50, or 60), `batch_k` (number of triplets per batch, set to the floor division of 300 and `batch_p` due to memory limits), `margin` (0.5, 1), `learning_rate` (0.0001, 1E-05, 5E-05, 6E-05, 7E-05, 8E-05, or 9E-05), `metric` (euclidean, squared euclidean, or cityblock), and `train_iterations` (10000, 15000, 20000, or 30000) parameters. A grid search was used to explore these hyperparameters.

Only a single triplet model was evaluated on the test dataset, and the images from the Schizophrenia Study were not used for model training.

## Embeddings

To perform downstream tasks, we averaged patch level embeddings to the level of images, donors, and genes. In the schizophrenia study, patch level embeddings were averaged within a specific brain individually for more granular analyses. We did not use individual patch level embeddings beyond visualization due to the computational costs of operating at this low level.

## Gene ontology annotations

The Gene Ontology (GO) was used as an additional source of gene classifications. This resource is much more general than the above annotations from Zeng and colleagues. Specifically, the GO consortium uses evidence from the biomedical literature to curate genes into groups. GO covers biology broadly, and while some annotations are brain-specific, most are not. GO groups with between 40 to 200 genes assayed in the Cortex Dataset were selected from all three ontologies (biological process, molecular function and cellular component), resulting in 55 groups.

## Experimental setup for annotation and diagnosis prediction

We used a simple setup to predict gene annotations in Cortex Study. We also used the similar setup for prediction of diagnosis for individuals in the Schizophrenia Study. Stratified five-fold cross-validation was used for all experiments. Each Zeng or GO annotation was predicted separately at the level of genes with a binary logistic regression classifier to accommodate genes with multiple annotations. No regularization penalty was used for the logistic regression, maximum iterations was limited to 500, and all other parameters were the default scikit-learn settings. We used AUC and F1 scores to gauge performance because most of the annotations from Zeng and GO are very imbalanced with limited positive examples.

For diagnosis prediction in the Schizophrenia Study, we also used a logistic regression classifier and a 5-fold cross-validation setup. Demographic and image embedding features were used for this task to make predictions for individual brains. Categorical demographic variables were encoded with 1-hot encoding. Like the gene-level annotation prediction, AUC and F1 were used as metrics.

## Source code and data availability

To ensure a consistent resource for further studies we have provided the code, supplementary tables, embeddings, and data files for reproducing the analyses publicly online at https://figshare.com/s/43ebba2711adc3ccdc13, and https://github.com/PegahA/Human_Brain_ISH_ML.

# Results

## Training on the cortex study dataset

The Cortex Study dataset assayed expression in the visual and temporal cortices in the neuro-pathologically normal human brains [4]. From this dataset, we used images from 42 donor brains (62% male) that assayed expression of 1,004 genes. Due to the limits of ISH techniques, not all genes were assayed in every brain. Specifically, each gene was assayed in 3.6 brains on average, but four genes were assayed in 41 or all of the donor brains (*CTNND2*, *GAP43*, *PCP4*, and *CARTPT*). On average, 84.9 genes were profiled per brain. Slightly more images were of

the temporal cortex (51%). Of the 23,258 total images, extraction of foreground patches was successful in all except 5. The result is a total of 11,155,856 256x256 image patches, providing the quantity needed to train and test a deep convolutional neural network.

The training and validation sets were used for tuning and hyper-parameter optimization. The rank-1 accuracy at the level of images was used to select the top model. As detailed in the Methods section, in the learned feature space, for all images, this measure is the proportion of images for which the closest image assays the expression of the same gene. We restricted this metric to images obtained from different brains to ensure that the network was not only learning to detect features specific to individual donors. Key parameters for the selected top model are the number of images per gene (`batch_k`) = 17, the number of genes per batch (`batch_p`) = 17, initial learning rate = 7e-5, augmentation through horizontal and vertical flips, the remaining variables were set their default values. In the training, validation, and test gene sets this final model rank-1 scores were 13.2%, 26.5%, and 38.3%, respectively. In comparison, features extracted from a Resnet50 trained to predict the class of the naturalistic images reached a rank-1 test set score of 24.9% (Table 1). The rank-1 accuracy progressively increases across the training, validation and test sets. This performance trend is the reverse of the expectation of an accuracy drop in unseen datasets. However, because we are testing gene re-identification, the number of unique genes in the splits partially determines the accuracy. For example, a training dataset of 80 genes with two images per gene would result in a random baseline rank-1 accuracy of 0.63% and a smaller test set of ten genes would reach 5.26%. To account for the number of unique genes in each set, we compare the metrics to the random baseline. As expected, this relative measure decreases from training to test (from 102 times the random performance to 13.2, Table 1).

## Learned embeddings predict expression level

Using annotations from the Cortex Study, we examined what the learned embeddings represent. At the level of images, the tissue source of temporal or visual cortex can be predicted from the embeddings from both the base ResNet (AUC = 0.884) and triplet loss embeddings (AUC = 0.902). Rank-1 scores calculated within the temporal (39.0%) and visual (34.6%) cortex images are not significantly different from the full result. Although limited to the temporal or visual cortices, this suggests information about the region assessed for expression is not a key indicator for gene re-identification. As described by Zeng and colleagues, each gene was manually characterized for expression intensity, laminar patterns, and cell-type specificity [4]. Expression intensity was graded on a range from—(no signal) to +++++ (over labeling). As seen in Fig 2, Uniform Manifold Approximation and Projection (UMAP) visualization of the gene level embeddings shows an expression intensity gradient [25]. Clustering of pattern and cell-type annotations are also apparent in this figure. Using supervised techniques, with the learned embeddings as input, expression intensity (average AUC = 0.898), pattern (average AUC = 0.879), and laminar specificity (average AUC = 0.805) were predicted with high levels of accuracy. In comparison, embeddings from the baseline ResNet trained on naturalistic images without gene re-identification training were broadly less accurate (average AUC's:

**Table 1. Rank-1 accuracy scores and fold increase over random in parentheses for each dataset split.**

| Model | Training | Validation | Test |
|---|---|---|---|
| **Random** | 0.129% (1x) | 0.851% (1x) | 2.90% (1x) |
| **ResNet** | 4.86% (37.7x) | 18.8% (22.1x) | 24.9% (8.57x) |
| **Triplet Loss** | 13.2% (102x) | 26.5% (31.1x) | 38.3% (13.2x) |

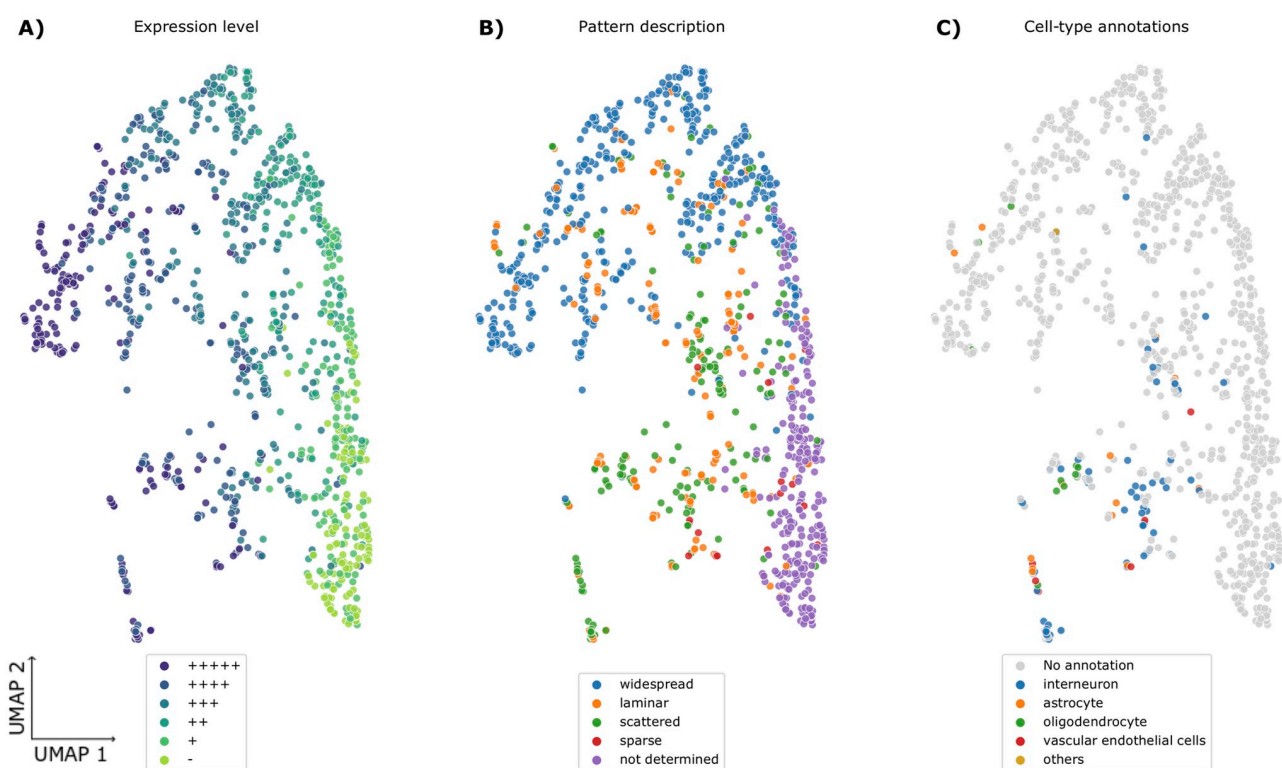

**Fig 2. Gene level embeddings represented in a 2-dimensional UMAP projection.** Genes are coloured to represent summary descriptions of ISH images by expert neuroanatomists as described in Zeng et al. (2012). Specifically, colours mark the level of expression in the V1 region and range from —(no signal) to +++++ (over labelling) (panel A). Panel B and C colour summarized expression pattern and cell-type specific expression, respectively.

expression intensity = 0.823, pattern = 0.802, and laminar specificity = 0.729). Similar differences are observed when comparing area under the precision-recall curve (AUC-PR) and the probability of the predictions exceeding chance in the test folds (maximum Mann–Whitney U test p-value across folds). Within the expression pattern annotations, the two largest sets that mark widespread and 'not determined' patterns were the best predicted (Table 2). The baseline ResNet embeddings predict 'not determined' annotations at levels nearing the trained triplet loss model. Similarly, the gap between the base ResNet and triplet loss models narrows for the lowest expression levels (S1 Table). This suggests that the baseline ResNet is best at representing images with low expression of the assayed gene. In summary, learned representations of the ISH images contain information about expression level and pattern.

**Table 2. Gene counts and prediction performance (AUC, AUC-PR, maximum p-value across folds) for Zeng et al. pattern description annotations.**

| Pattern | # Genes | ResNet Base | | | Triplet | | |
|---|---|---|---|---|---|---|---|
| | | AUC | AUC-PR | Max p-value | AUC | AUC-PR | Max p-value |
| widespread | 417 | 0.893 | 0.855 | 1.13E-17 | 0.943 | 0.931 | 2.72E-24 |
| sparse | 16 | 0.655 | 0.091 | 5.85E-01 | 0.713 | 0.092 | 8.83E-01 |
| scattered | 151 | 0.819 | 0.541 | 3.07E-07 | 0.896 | 0.697 | 6.40E-10 |
| not determined | 263 | 0.911 | 0.769 | 5.38E-18 | 0.969 | 0.917 | 1.01E-20 |
| laminar | 125 | 0.731 | 0.336 | 5.27E-03 | 0.877 | 0.594 | 3.86E-07 |

## Performance on GO membership prediction

We next tested if the learned embeddings can predict a broader range of gene annotations. Compared to the manual annotations used above, predicting GO membership is a harder task because they are derived from the literature and not specific to the brain [26]. Across the 55 tested GO groups, the average AUC was 0.60 (full listing in S2 Table). The most accurately predicted GO group contains the 85 genes annotated to 'G protein-coupled receptor activity' (AUC = 0.839), followed by 'chloride transmembrane transport'(AUC = 0.756). Surprisingly, embeddings from the ResNet trained on naturalistic images alone reached slightly higher AUC values (average AUC 0.62) with the G protein group top-ranked (AUC = 0.858). It has been previously noted that G protein-coupled receptor genes are expressed at low levels in the Cortex Study [4]. Upon inspection of the other top GO groups, it appears that this low or lack of expression may be the main feature used to classify genes in this analysis. For example, the 'chloride transmembrane transport' group has the highest proportion of genes marked as having no expression (31%). We also suspected that the assayed genes were not randomly distributed across donors. To test this, we constructed a donor embedding matrix where each gene is represented by a binary vector that identifies which donors that gene was assayed in. Predicting GO membership with this control binary matrix results in an average AUC of 0.673, surpassing the base ResNet and triplet learned embeddings. The top-scoring GO group is again 'G protein-coupled receptor activity'(AUC = 0.923). As expected, the per donor proportion of these genes assayed varies greatly. For example, 41% of the genes assayed in a donor H08-0002 are members of this group, while the average across all donors is 4.5%. This suggests that the ResNet embeddings and, to a lesser degree, the triplet embeddings contain donor specific signals that allow prediction of this and other GO groups. In contrast, the donor embedding matrix was not able to predict the annotations from Zeng and colleagues (average AUC's: expression intensity = 0.53, pattern = 0.53, and laminar specificity = 0.46). While GO group membership can be predicted from the learned embeddings, this task is not useful for evaluating the learned representations due to donor-specific bias.

## Generalization to unseen ISH images

Using the trained model, we generated and tested embeddings of an unseen set of 8,347 images from a study of the dorsolateral prefrontal cortex. This study assayed postmortem expression in 20 individuals with schizophrenia and 33 control individuals [7]. Compared to the Cortex Study, these images are from different brains and cortical region (dorsolateral prefrontal cortex). However, we note that both studies used the same ISH platform, and some genes are assayed in both the Cortex and Schizophrenia studies (73 of 78 genes assayed). Using the model trained on all genes assayed in the Cortex Study dataset, rank-1 from the resulting embeddings reaches 60.2%. This exceeds our previous rank-1 values due to the limited set of 78 genes assayed in this study. Performance of random and ResNet embeddings are much lower at 1.56% and 21.8%, respectively. Using a model that was not trained on the overlapping genes results in a much lower rank-1 score of 42.2%, suggesting distinct features of the 73 overlapping genes are learned during training. Overall, the model's gene re-identification ability is impressive for both novel and trained genes in this new dataset.

## Detection of altered gene expression in individuals with schizophrenia

We next used the model-generated embeddings to test for altered expression that is associated with schizophrenia. Using embeddings at the level of genes and individuals, we attempted to predict schizophrenia diagnosis. Using a set of 20 genes that were assayed in all 53 individuals,

the average embedding of these genes could not significantly predict schizophrenia diagnosis (AUC = 0.59). This suggests there is not a broad disruption of expression patterns.

Using triplet embeddings for the 62 genes assayed in 50 or more individuals, the average AUC for diagnosis prediction was 0.58. For comparison, random embeddings reach a top AUC of 0.72 and have an average AUC of 0.50 across all genes. For all except three genes, embeddings from the base ResNet trained on naturalistic images underperform embeddings obtained from training on the Cortex Study (based on AUC metric). In aggregate, the average AUC metric for the base ResNet is much lower than the triplet embeddings (0.39 vs 0.58). Surprisingly, the base ResNet AUC values are lower than random and our learned embeddings (paired t-test, p $< 10^{-7}$). In contrast, the triplet embeddings AUC scores are significantly better than those random embeddings (paired t-test, p $<$ 0.0005). Using the F1 metric, the base ResNet values are not different from random (paired t-test, p = 0.13), while the triplet embeddings again outperform classifiers using random embeddings (paired t-test, p $< 10^{-4}$). Overall, across all genes, embeddings from a base ResNet perform worse or at par to random embeddings while the learned triplet embeddings are more accurate.

The top gene in the triplet analyses, Sodium Voltage-Gated Channel Beta Subunit 4 (*SCN4B*), reaches an AUC score of 0.83 and F1 of 0.63 (top ten genes are listed in Table 3). To visualize *SCN4B* patterns that are most informative, we selected the embedding dimension that was most predictive of diagnosis (post hoc, dimension 40). Of the 7,050 patches of *SCN4B* expression, we examined the five patches with the lowest and highest values for this embedding dimension. Across these ten most extreme patches, each is from a different brain, suggesting it is not a donor-specific signal. Fig 3 shows that the highest activating patches show intense laminar and scattered labelling, while the patches with the lowest activation have lower expression intensities in fewer cells. In agreement, the Cortex Study annotated this gene as having specific expression in layers 3 and 5 [4].

Due to the challenges of postmortem brain studies, some experimental factors and demographic variables are not well balanced between controls and the individuals diagnosed with schizophrenia. Specifically, tissue pH, age, and smoking status are significantly different between the two groups [7]. Using the demographic variables as input variables instead of image embeddings results in an AUC of 0.857 and an F1 score of 0.71 (52 donors). The embeddings of only *CUX2* exceed the demographic predictor performance when tested in the same set of individuals (AUC of 0.78 vs 0.76, n = 53). By combining the demographic variables with the learned embeddings, we identify only 5 genes that surpass the performance of the demographic-based classifier (*CUX2*, *CNP*, *GAD2*, *GRIK1*, and *SLC17A7*). These genes provide only a marginal performance improvement, suggesting the expression patterns that discriminate individuals with schizophrenia mostly reflect differences due to age, pH or smoking status.

**Table 3. AUC scores for the top ten genes most predictive of schizophrenia diagnosis.**

| Gene Symbol | Number of Donors | Embeddings | Demographic variables | Demographics + embeddings |
|---|---|---|---|---|
| SCN4B | 52 | 0.835 | 0.857 | 0.832 |
| BDNF | 51 | 0.829 | 0.840 | 0.739 |
| CUX2 | 53 | 0.779 | 0.763 | 0.793 |
| RASGRF2 | 53 | 0.750 | 0.763 | 0.751 |
| CIT | 53 | 0.735 | 0.763 | 0.726 |
| TRMT9B | 50 | 0.734 | 0.773 | 0.654 |
| PVALB | 53 | 0.720 | 0.763 | 0.727 |
| PPP1R1B | 52 | 0.712 | 0.832 | 0.750 |
| GAD2 | 53 | 0.702 | 0.763 | 0.801 |
| CTGF | 52 | 0.696 | 0.760 | 0.726 |

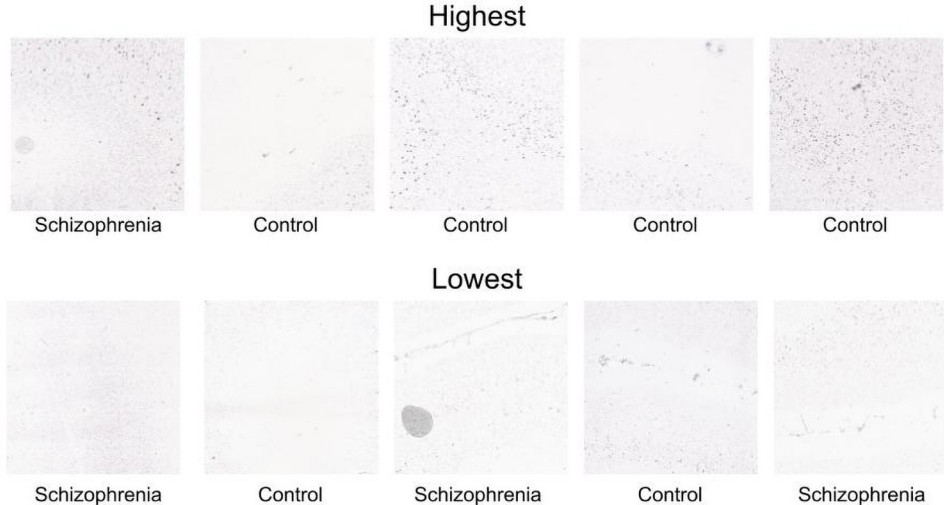

**Fig 3. Patches of *SCN4B* ISH expression images that most (top) and least (bottom) activate embedding dimension 40.** Each patch is from a unique donor, and the diagnostic group is labelled for each.

## Discussion

In this paper, we evaluate gene re-identification as a machine learning approach to generate gene expression representations. In comparison to the input images, the learned representations are compact and transfer to downstream tasks. We show that these embeddings contain biologically meaningful information. For example, they can be used to predict gene function and expression patterns. Importantly, we also show that the learned neural network model generalizes to images from a different cortical region. Furthermore, the extracted embeddings highlight genes that may be disrupted in schizophrenia.

Although the genes assayed by Guillozet-Bongaarts et al. are enriched for previous associations with schizophrenia we discuss the top three genes from our analyses. Our top-ranked gene, Sodium Voltage-Gated Channel Beta Subunit 4 (*SCN4B*), has not been associated with schizophrenia, but voltage-gated sodium channels have broadly been implicated in its pathogenesis [27]. Also, mutations in this gene cause long QT syndrome [28]. Prolonged QT has been observed in drug-free patients with schizophrenia and is a side-effect of antipsychotic drugs [29, 30]. The second-ranked gene, brain-derived neurotrophic factor (*BDNF*) has been implicated in schizophrenia but human studies have not clearly associated it with the disorder [31]. Ranked third, cut like homeobox 2 (*CUX2*) is the only gene where the embeddings improve performance when added to the demographic information. *CUX2* is a marker of layer 2/3 excitatory neurons and is involved in dendritogenesis [32, 33]. No genetic or molecular evidence has linked this gene to schizophrenia, but differences in its expression pattern may indicate abnormal dendrites or neurodevelopmental processes. No strong differences in cell density or staining intensity were found for *BDNF*, *SCN4B* and *CUX2* in the analyses of Guillozet-Bongaarts and colleagues. However, *TRMT9B* (*C8orf79* or *KIAA1456*), which was ranked sixth in our analyses, was highlighted in their abstract [7]. While our study highlights genes that may inform schizophrenia pathology, their differences in expression may be due to demographic or antipsychotic treatment.

To our knowledge, our approach is the most computationally intensive analysis of human brain ISH images to date. Specifically, previous analyses of the Schizophrenia Study were limited to manually determined regions of interest patches. These patches were assessed with a

hand-engineered informatics pipeline that extracted cell density and staining intensity [7]. In contrast, our model represents 50 patches per image with 128 values. Our current approach that averages these patches together might be less sensitive to larger spatial signals and dilute localized abnormalities. Utilizing the full set of patch embeddings for downstream tasks could address this limitation. In comparison, Guillozet-Bongaarts and colleagues' analyses provide interpretability as they localize expression alterations to specific cortical layers and prefrontal regions. Application of techniques for interpretability in computational pathology may allow localization of key features, but several challenges remain [34].

Our approach has several limitations that motivate future research in this area. In the context of predicting diagnoses, we note that schizophrenia is a very heterogeneous disorder [35]. The difficulty of identifying schizophrenia from brain images evident from a recent MRI study that reported an AUC of 0.71 in a held-out test dataset [36]. Regarding scale, our input patches of 65 thousand pixels capture less than 1% of the original image resolution. As deep learning tools improve, we aim to test our approach on finer images. We also note that the input datasets are limited to approximately 1,000 genes, a small fraction of the over 20,000 protein-coding genes in the human genome. Other datasets of ISH images from children with autism and the aged brain provide additional sources but are limited in the number of genes assayed [37, 38]. To provide a genome-wide perspective, the Allen Mouse Brain Atlas could serve as an additional pre-training dataset for gene re-identification even though species-specific expression patterns are common [4, 39]. Unlike person re-identification, gene re-identification is not the primary goal. Our transfer learning approach tests the image embeddings for downstream tasks separately. An alternative is to develop an end-to-end model that learns to optimize a composite score that weights various tasks. This multi-task learning approach has been used successfully to encode sentences [40]. Our goal was to generate universal embeddings that are useful for supervised and unsupervised tasks. If a clear classification target is available then the gene re-identification approach might not outperform a more direct end-end model. However, pre-training on gene re-identification may provide a strong initial model when existing pre-trained models are not applicable such as a new architecture or data type.

## Conclusions

In summary, we introduce gene re-identification as a novel contrastive learning task. Using the assayed gene's identity as a label, we leverage large gene expression datasets to learn robust representations of gene expression images. Representations learned from ISH images of the human neocortex carry information about gene function and expression patterns. Furthermore, our approach highlights genes of interest that have altered embeddings in images from individuals diagnosed with schizophrenia. The datasets we use are open, and we have publicly released our source code, learned embeddings, and models to spur research in this area.

## Supporting information

**S1 Table. Gene counts and prediction performance statistics for all Zeng et al. annotations.**
(XLSX)

**S2 Table. Gene counts and prediction performance statistics for all tested gene ontology annotations.**
(XLSX)

## Acknowledgments

We thank the Allen Institute for Brain Science and their collaborators for making the ISH study data available. We thank Sean Hill for insightful comments and suggestions. We thank Dessa, Dell, and NVIDIA for machine learning software and hardware support.

## Author Contributions

**Conceptualization:** Pegah Abed-Esfahani, Benjamin C. Darwin, Jason Lerch, Leon French.

**Data curation:** Pegah Abed-Esfahani, Derek Howard, Nick Wang, Ethan Kim.

**Formal analysis:** Pegah Abed-Esfahani, Benjamin C. Darwin, Derek Howard, Leon French.

**Funding acquisition:** Leon French.

**Investigation:** Leon French.

**Methodology:** Pegah Abed-Esfahani, Benjamin C. Darwin, Derek Howard, Nick Wang, Ethan Kim, Jason Lerch, Leon French.

**Project administration:** Jason Lerch, Leon French.

**Resources:** Pegah Abed-Esfahani, Jason Lerch, Leon French.

**Software:** Pegah Abed-Esfahani, Derek Howard, Leon French.

**Supervision:** Jason Lerch, Leon French.

**Validation:** Leon French.

**Visualization:** Pegah Abed-Esfahani, Derek Howard, Leon French.

**Writing – original draft:** Pegah Abed-Esfahani, Benjamin C. Darwin, Leon French.

**Writing – review & editing:** Pegah Abed-Esfahani, Benjamin C. Darwin, Derek Howard, Nick Wang, Ethan Kim, Jason Lerch, Leon French.

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
