## [Decision Letter · Decision Letter 0]

16 Dec 2021

PONE-D-21-23122

Evaluation of deep convolutional neural networks for in situ hybridization gene expression image representation

PLOS ONE

Dear Dr. French,

Thank you for submitting your manuscript to PLOS ONE. After careful consideration by a Reviewer and an Academic Editor, all of the critiques of Reviewer #1 must be addressed in detail in a revision to determine publication status. If you are prepared to undertake the work required, I would be pleased to reconsider my decision, but revision of the original submission without directly addressing the critiques of the Reviewer does not guarantee acceptance for publication in PLOS ONE. If the authors do not feel that the queries can be addressed, please consider submitting to another publication medium. A revised submission will be sent out for re-review. The authors are urged to have the manuscript given a hard copyedit for syntax and grammar.

**Comments to the Author**

1. Is the manuscript technically sound, and do the data support the conclusions?

Reviewer #1: Partly

2. Has the statistical analysis been performed appropriately and rigorously? 

Reviewer #1: Yes

3. Have the authors made all data underlying the findings in their manuscript fully available?

Reviewer #1: Yes

4. Is the manuscript presented in an intelligible fashion and written in standard English?

Reviewer #1: Yes

5. Review Comments to the Author

Reviewer #1: The author proposed gene re-identification as a ML approach to represent ISH gene expression images.

The topic is of interest; however, the authors can improve the description of the work to emphasize why this approach constitutes a consistent resource for the researcher in this field (especially in the introduction/methods description).

Also it is not totally clear why the performance on the test set is always higher than the validation/training set. Does it depend on a not optimal split ratio? Did the authors properly perform some approaches, for instance, cross-validation to prevent such a problem?

Furthermore, the authors could define the range of possible values for all hyperparameters tested for tuning the network. Also, the authors should add the conclusion section.

minor:

- the abbreviations should be defined when they first appear in the text (e.g. SCN4B in the Abstract)

- Images should stay close to their respective captions.

6. PLOS authors have the option to publish the peer review history of their article (what does this mean?). If published, this will include your full peer review and any attached files.

**Do you want your identity to be public for this peer review?** For information about this choice, including consent withdrawal, please see our Privacy Policy.

Reviewer #1: No

We look forward to receiving your revised manuscript.

Kind regards,

Stephen D. Ginsberg, Ph.D.

Section Editor

PLOS ONE

“This study was supported by the CAMH Foundation, McLaughlin Centre, Canada Foundation for Innovation, and a National Science and Engineering Research Council of Canada (NSERC) Discovery Grants to LF.”

“This study was supported by the CAMH Foundation, McLaughlin Centre, Canada Foundation for Innovation, and a National Science and Engineering Research Council of Canada (NSERC) Discovery Grant.”

“This study was supported by the CAMH Foundation, McLaughlin Centre, Canada Foundation for Innovation, and a National Science and Engineering Research Council of Canada (NSERC) Discovery Grants to LF.”

---

## [Author Response · Author response to Decision Letter 0]

23 Dec 2021

> Reviewer #1: The author proposed gene re-identification as a ML approach

> to represent ISH gene expression images.

> 

> The topic is of interest; however, the authors can improve the

> description of the work to emphasize why this approach constitutes a

> consistent resource for the researcher in this field (especially in the

> introduction/methods description).

We agree and have included this paragraph to the introduction:

“Our approach and tools to undertake gene re-identification establish a consistent resource for neuroscience and machine learning researchers. In comparison to person re-identification tasks and datasets that raise privacy concerns (Dietlmeier et al. 2020), images of anonymized brain tissue samples cannot be used to identify subjects. Furthermore, unlike most entities used in re-identification studies, genes are relatively static and allow generalization tests. These attributes provide a stable framework to assess new techniques and discover insights into biological processes.”

We also now mention consistency in the Methods section:

“To ensure a consistent resource for further studies we have provided the code, supplementary tables, embeddings, and data files for reproducing the analyses publicly online at https://figshare.com/s/43ebba2711adc3ccdc13, and https://github.com/PegahA/Human_Brain_ISH_ML.”

> Also it is not totally clear why the performance on the test set is

> always higher than the validation/training set. Does it depend on a not

> optimal split ratio? Did the authors properly perform some approaches,

> for instance, cross-validation to prevent such a problem?

Yes, we are confident in our approaches. The large dataset size, which contains 23,258 total images and over 11 million image patches, removes the need for cross-validation. In agreement, the 80/10/10 split we used is typical for neural network experiments (for example, it is the default for Google AutoML: https://cloud.google.com/automl-tables/docs/prepare).

To clarify the high test set performance, please consider an example dataset of 100 genes with two images per gene, resulting in 200 total images. Using the same 80/10/10 training/test/validation split and generating random embeddings, we would have the following results:

Training set:

80 genes, two images per gene to provide 160 total images. Given one of these images, the chance of the paired image of the same gene being closest in the random embedding space is 1/159 = 0.63%

Validation set:

10 genes, two images each - 20 images, expected random rank-1 accuracy is 1/19 = 5.26%

Test set:

10 genes, two images each - 20 images, expected random rank-1 accuracy is 1/19 = 5.26%

Here the accuracy in the validation and test sets is 8.35 times that of the training set without any learning taking place. This demonstrates that the rank-1 metric needs to be considered in the context of the evaluation set size (number of unique genes). We have added this multiple over random embeddings to our results to show our rank-1 accuracy decreases in the validation and tests sets as compared to a random baseline (Table 1). Specifically, Triplet Loss is 102x more accurate than random embeddings in training, which drops to 31x and 13x in the validation and test sets, respectively. This agrees with the accuracy reductions that are expected in new examples. Our tests further validate these results in the completely unseen dataset obtained from the dorsolateral prefrontal cortex (38x random rank-1 accuracy). These results on unseen data allow us to dismiss any concerns of incorrect training of the model.

We have added the following text to provide a shortened version of the above example to the readers to prevent confusion:

“For example, a training dataset of 80 genes with two images per gene would result in a random baseline rank-1 accuracy of 0.63% and a smaller test set of ten genes would reach 5.26%. To account for the number of unique genes in each set, we compare the metrics to the random baseline. As expected, this relative measure decreases from training to test (from 102 times the random performance to 13.2, Table 1).”

> Furthermore, the authors could define the range of possible values for

> all hyperparameters tested for tuning the network. 

We agree and we now list the hyperparameters tested in the Methods section:

“When tuning the hyperparameters on the training and validation sets, we experimented with different flip_augment (true or false), batch_p (number of genes per batch: 5, 10, 12, 14, 16, 17,18, 20, 40, 50, or 60), batch_k (number of triplets per batch, set to the floor division of 300 and batch_p due to memory limits), margin (0.5, 1), learning_rate (0.0001, 1E-05, 5E-05, 6E-05, 7E-05, 8E-05, or 9E-05), metric (euclidean, squared euclidean, or cityblock), and train_iterations (10000, 15000, 20000, or 30000) parameters. A grid search was used to explore these hyperparameters.”

> Also, the authors should add the conclusion section.

We agree and now mark our concluding Discussion paragraph as our Conclusion. 

> minor:

> - the abbreviations should be defined when they first appear in the text

> (e.g. SCN4B in the Abstract)

Thank you for pointing this out, we now provide the full name of SCN4B in the abstract and in the main text.

> - Images should stay close to their respective captions.

In the PLOS One manuscript submission system, the images are not provided inline. All figures should appear at the end of the manuscript file, if not, you may be reviewing the wrong file. Currently, the figure titles and captions appear directly after the paragraph they are mentioned as per the PLOS One guidelines. We will ensure that our final version has properly placed figures.

---

## [Editor Report · Decision Letter 1]

3 Jan 2022

Evaluation of deep convolutional neural networks for in situ hybridization gene expression image representation

PONE-D-21-23122R1

Dear Dr. French,

We’re pleased to inform you that your manuscript has been judged scientifically suitable for publication and will be formally accepted for publication once it meets all outstanding technical requirements.

Kind regards,

Stephen D. Ginsberg, Ph.D.

Section Editor

PLOS ONE

---

## [Editor Report · Acceptance letter]

13 Jan 2022

PONE-D-21-23122R1 

Evaluation of deep convolutional neural networks for in situ hybridization gene expression image representation 

Dear Dr. French:

I'm pleased to inform you that your manuscript has been deemed suitable for publication in PLOS ONE. Congratulations! Your manuscript is now with our production department. 

Kind regards, 

on behalf of

Dr. Stephen D. Ginsberg 

Section Editor

PLOS ONE